# Modeling the Angle-Specific Isokinetic Hamstring to Quadriceps Ratio Using Multilevel Generalized Additive Models

**DOI:** 10.3390/medicina55080411

**Published:** 2019-07-26

**Authors:** Lucas A. Sousa, André L. A. Soares, Ahlan B. Lima, Roberto R. Paes, Luiz R. Nakamura, Humberto M. Carvalho

**Affiliations:** 1School of Physical Education, University of Campinas, Campinas, São Paulo 13083-851, Brazil; 2Department of Physical Education, School of Sports, Federal University of Santa Catarina, Florianópolis, Santa Catarina 88040-900, Brazil; 3Department of Informatics and Statistics, School of Technology, Federal University of Santa Catarina, Florianópolis, Santa Catarina 88040-900, Brazil

**Keywords:** isokinetic, muscle function, strength, adolescent, Bayesian methods

## Abstract

*Background and Objectives:* This study considered the use of a generalized additive multilevel model to describe the joint-angle-specific functional hamstring to quadriceps ratio (H:Q ratio) in the knee, using all of the available truly isokinetic data within the range. *Materials and Methods:* Thirty healthy male basketball players aged 15.0 (1.4) years (average stature = 180.0 cm, *SD* = 11.1 cm; average body mass = 71.2, *SD* = 14.9 kg) years were considered. All players considered had no history of lower extremity musculoskeletal injury at the time of testing or during the 6 months before testing, and had been engaged in formal basketball training and competition for 5.9 (2.4) years. Moments of force of the reciprocal concentric and eccentric muscular actions for the knee extensors and flexors assessed by isokinetic dynamometry at 60°∙s^−1^ were used. *Results:* Maximum moments of force were attained at different angle positions for knee extension. For knee flexion, it was apparent that there was an ability to maintain high levels of moment of force between 30° and 60° in the concentric muscular action, corresponding to the concentric action of the hamstrings. However, for the eccentric knee flexion, corresponding to the quadriceps action, there was a marked peak of moment of force at about 55°. The functional H:Q ratio for the knee extension was non-linear, remaining higher than 1.0 (i.e., point of equality) from the beginning of the extension until approximately 40° of the knee extension, leveling off below the point of equality thereafter. On average, the functional H:Q ratio for the knee flexion did not attain 1.0 across the range of motion. The functional H:Q ratio for the knee in the present sample peaked at 20° and 80°, declining between these angle positions to below 0.50 at about 0.54. *Conclusions:* Estimating the form of the non-linear relationship on-the-fly using a generalized additive multilevel model provides joint-angle-specific curves and joint-angle-specific functional H:Q ratio patterns, allowing the identification and monitoring of strength development, with potential implications for injury and performance.

## 1. Introduction

Hamstrings and knee injuries, especially non-contact anterior cruciate ligament tears, are two of the most common severe injuries in sports [1,2]. In particular, basketball effort demands involve activities with a repeated high intensity of stretch–shortening cycles, including jumping or cutting typical in basketball, football, or volleyball, amongst other sports. There has been an increased awareness about the incidence of knee injuries among young athletes [3,4]. Children and adolescents in sports are exposed to situations ranging from informal play to systematic training and competition, hence underlining the need for comprehensive and authoritative information on the risk and protective factors.

Isokinetic strength assessment provides information about the maximal dynamic muscular action when the velocity of the movement is controlled and kept constant [5,6]. Isokinetic assessment has been favored to monitor muscle strength and balance involved in knee stability in applied clinical contexts and research [5,7,8,9,10,11]. Within sports contexts, isokinetic assessment is an important aspect of training and rehabilitation for the prevention of serious sport-related injuries [11,12].

Isokinetic strength assessment interpretations are mainly based on the single point peak moment. The balance of strength among muscles spanning the knee is typically described by the hamstring/quadriceps peak moment ratio [10]. Traditionally the hamstring/quadriceps (H:Q) peak moment ratios are based on peak moments during maximal voluntary concentric actions [13]. However, it has been noted that the agonist–antagonist strength relationship for knee extension and flexion may be better described by a functional H:Q [9]. In this case, a ratio of eccentric hamstring to concentric quadriceps muscle strength is representative of knee extension, and a ratio of concentric hamstring to eccentric quadriceps muscle strength is representative of knee flexion. Although the functional H:Q ratios account for the role of the antagonist in knee joint stabilization at specific joint angles, functional H:Q ratios fail to account for the hamstring–quadriceps relationship throughout the entire range of motion [10]. Generally, joint-angle-specific functional H:Q ratios have been determined using a single angle-specific moment of force values of reciprocal concentric and eccentric muscle actions [10,14,15,16]. Moreover, visual or repeated-measures analysis of variance-based extrapolation are often considered within the observed range of motion [17], which may lead to unrealistic prediction values.

The use of traditional repeated-measures analysis of variance-based approaches is limited to describing joint-angle-specific curves, and joint-angle-specific functional H:Q ratios in particular. Traditional analysis of variance-based approaches disregard the violation of assumptions that likely occurs due to temporal dependencies in the data or that responses constitute time series, which raises the problem of autocorrelated errors [18]. Furthermore, isokinetic assessment data present a longitudinal structure, where consecutive moments of force observed across the range of motion are nested within each individual. Hence, this hierarchical structure needs to be modeled appropriately to describe variation within and between individuals, allowing for a consideration of the covariates between individuals or to explore how much of the between-individual variation can be explained by other factors [19].

Generalized additive models (gam) are popular, powerful, and flexible modeling functions used to estimate smooth and non-linear trends in time series [20,21,22]. On the other hand, multilevel models, also known as hierarchical models or mixed effects models, can explicitly consider clusters of observation within a set of data, such as repeated measures within an athlete that have unique coefficients and allow the data to be related by simultaneously modeling the population of clusters [23,24]. The combination of gam and hierarchical models into one framework, albeit complex [25], allows the functional relations predictor to be smoothed and response to vary between individuals or groups in a way where the described functions are pooled toward a common shape [26].

The ability to estimate the form of the non-linear relationship on-the-fly using a generalized additive multilevel model (gamm) [27,28] is very appealing for describing joint-angle-specific curves, and joint-angle-specific functional H:Q ratios in particular, potentially allowing the identification and monitoring of patterns of strength development, with implications for injury and performance [29]. On the other hand, it allows all the available data in the observed range of motion to be modeled and between individual variation to be inspected altogether. The purpose of the present article is to illustrate the use of gamm to describe joint-angle-specific functional H:Q ratios among a sample of adolescent basketball players.

## 2. Materials and Methods

### 2.1. Sample

The sample included 30 healthy male basketball players aged 15.0 (1.4) years (average stature = 180.0 cm, *SD* = 11.1 cm; average body mass = 71.2, *SD* = 14.9 kg). On average, players had been engaged in formal basketball training and competition for 5.9 years (2.4), with experience in strength training for at least one year. The players were from the youth basketball program from a club in the Campinas metropolitan region, São Paulo State, Brazil, and competed at a state level supervised by the Federação Paulista de Basketball (FPB). The study was approved by the Research Ethics Committee of the University of Campinas (nº 49143515.3.0000.5404, 19/11/2015) and was conducted in accordance with recognized ethical standards [30]. Participants were informed about the nature of the study and also that participation was voluntary and that they could withdraw from the study at any time. Participants and their parents or legal guardians provided informed written consent. No player was suffering from lower extremity musculoskeletal injury at the time of testing or during the 6 months before testing that limited activity for more than 48 h.

### 2.2. Isokinetic Dynamometry Assessment

We considered the reciprocal concentric muscular actions of knee flexion and extension at an angular velocity of 60°∙s^−1^ of the adolescent male basketball players. Details of the isokinetic measurements and reliability estimates of the observer were reported elsewhere [31]. Briefly, isokinetic assessments of reciprocal knee extension and flexion muscular actions were made using a calibrated dynamometer (Biodex System 3, Shirley, NY, USA). All players performed a standardized warm-up, where each player performed a 10 min cycling warm-up on a cycle ergometer with minimal resistance at 60 rev min^−1^, and 2 min of static stretching of the hamstring and quadricep muscles. After the warm-up, the athlete was placed in a seated position adjusted according to manufacturer guidelines in a standardized 85° hip flexion from the anatomical position. Only reciprocal muscular actions in the dominant leg were considered here. The lever arm of the dynamometer was aligned with the lateral epicondyle of the knee and the force pad was placed approximately 3 to 5 cm superior to the medial malleolus, with the ankle in a plantigrade position. Range of motion during testing was set using voluntary maximal full extension ranging from 0° to 90° of knee flexion. Cushioning was set using a hard deceleration (according to manufacturer guidelines) and 90° thus constituted the range of motion tested. Effects of gravity on the limb and lever arm were accounted for. Hands were placed in the hand grips at the sides of the Biodex System 3 seat during the test procedure. Each adolescent player performed five continuous maximal repetitions. Visual feedback of moment versus time was provided during the test, but no verbal feedback was given [32].

### 2.3. Measures

Joint-angle-specific curves for the concentric knee extension (quadriceps) and flexion (hamstrings), and eccentric knee extension (hamstrings) and flexion (quadriceps), were based on maximal knee moments of force at each angular position from the best repetitions, expressed as N·m, and from only moments of force that were “truly” isokinetic at 60°∙s^−1^. The functional H:Q ratio for the knee extension using eccentric hamstrings by concentric quadriceps moments of force at each corresponding angle position, and the functional H:Q ratio for the knee flexion using concentric hamstrings by eccentric quadriceps moments of force at each corresponding angle position, were determined. A point of equality, i.e., a functional H:Q ratio of 1.0, may be interpreted, for example, for the knee extension, as the eccentrically acting hamstrings have the ability to fully brake the action of the concentrically contracting quadriceps [10].

### 2.4. Statistical Analysis

We considered each moment of force measurement across the range as repeated observations over time, in this case, over successive angular positions (at level 1) nested within individuals (level 2), representing a time series. To deal with the hierarchical structure of the data and expectable non-linear trends [29,33], we used gamm to fit smooth terms to describe isokinetic strength curves. We considered a single common smoother, where all individuals had a similar functional response, but allowed for variation between individuals in the response, i.e., between-individual variation in the intercept and smooth terms were allowed in the models. The models here may be considered a close analogue to a multilevel model with varying slopes [26]. The gamm models were estimated using a fully-Bayesian approach via the brm() function available in the brms package, available as a package in the R statistical language, using the default priors, i.e., uninformative [34]. A main advantage of gamm is that it allows for smooth trends to be estimated from the data itself. Comparisons of competing models were made with the widely applicable information criteria (WAIC) [24,35].

## 3. Results

The codes and summary of all the models describing knee extension and flexion in both concentric and eccentric muscular actions, as well as the functional H:Q ratio for the knee extension and knee flexion, are summarized as Appendix A. In all models, the credible intervals of the standard deviations of the smooth coefficients were sufficiently far away from zero (as visible under ‘Smooth Terms’ in the summary of each model), indicating the non-linearity response of moments of force across the range of motion. On the other hand, for all models, both intercepts and smooth terms varied considerably by individuals (as visible under ‘Group-Level Effects’ in the summary of each model).

The posterior predictions for knee extension and flexion in both concentric and eccentric muscular actions are displayed in Figure 1. Note that angle positions were rescaled for knee extension in order to provide 0° at the start of the action (i.e., starting position corresponds to a knee flexion at 90°). By inspecting both panels A and B in Figure 1, it is clear that maximum moments of force were attained at different angle positions for knee extension. As for knee flexion (Figure 1, panels C and D), it was apparent that there was an ability to maintain high levels of moment of force between 30° and 60° in the concentric muscular action, corresponding to the concentric action of the hamstrings. However, for the eccentric knee flexion, corresponding to the quadriceps action, there was a marked peak of moment of force at about 55°.

The posterior predictions for the functional H:Q ratio for the knee extension (panel A) and the knee flexion (panel B) are displayed in Figure 2. The functional H:Q ratio for the knee extension was non-linear, remaining higher than 1.0 (i.e., point of equality) from the beginning of the extension until approximately 40° of the knee extension, leveling off below the point of equality thereafter. On average, the functional H:Q ratio for the knee flexion presented did not attain 1.0 across the range of motion. The functional H:Q ratio for the knee in the present sample peaked at 20° and 80°, lowering between these angle positions to below 0.50 at about 0.54.

## 4. Discussion

Physiological functions during exercise are all conceptualized in respect to time [36]. However, it is often conventional to reduce them to a single estimate when describing physiological functions, such as the force-length properties of a whole group of synergetic muscles. There have been exciting advances lately in the analytical approaches available to model complex time-dependent variables, given the advance of computational capabilities and their availability to the end-user researcher. Hence, in this paper, we intended to illustrate an applied example, describing knee joint-angle-specific functional H:Q ratios using gamm.

The interpretation of knee joint-angle-specific functional H:Q ratios is very important to understand the dynamic knee joint stability [16,17]. Recently, the interest in joint-angle-specific functional H:Q ratios has increased substantially [12,14,15,16,17,37,38,39]. Joint-angle-specific functional H:Q ratios have mostly been determined based on single angle-specific moments of force (e.g., 15°, 30°, and 45°) [11,15,16,37,39], or interpolating polynomial models using moment of force values for knee-joint angles within an interval (e.g., 5° intervals) [14]. These procedures are time consuming, and potentially useful data and variation may be missed. Another limitation when considering joint-angle-specific isokinetic data lies in the disregard of the structure of the hierarchical data. Each angle-specific moment of force measurement is nested within each individual. Hence, there is the need to appropriately model variation within and between individuals across the range of observation. Using a multilevel modeling framework allows researchers to explicitly overcome this limitation [19,23,29]. Furthermore, an immediate advantage of using gamm is the ability to estimate the form of the non-linear joint angle-specific moment of force, or the joint angle-specific functional H:Q ratios across the range of motion. The interpretation of the shapes is relevant to interpretations of individual performances, with implications for injury [29].

Considering the joint angle-specific moment of force of the knee extension and flexion (both concentric and eccentric muscular actions), in the present sample, it showed that maximum moments of force were attained at different angle positions (Figure 1). The shapes of the joint angle-specific moment of force for the knee extension and flexions considering concentric muscular actions were similar to the limited available data [5,29,40,41]. Differences in moments of force and angle at maximum moments may vary due to differences between samples, such as age, training status, and occurrence of knee injury. On the other hand, the models in the present study are consistent with the original observations proposing the functional H:Q ratios [8,9]. Overall, there is a need to consider joint angle-specific functional H:Q ratios across the range of motion to appropriately interpret the dynamic knee joint stability. 

In the present study, the functional H:Q ratio for the knee extension was substantially higher than 1.0 values from the start of the extension. The functional H:Q ratio for the knee extension decreased with the increase of the range of motion of the knee extension, leveling off below the point of equality at about 40° of the knee extension. This is attributed to the lower eccentric moment of force production of the hamstrings compared to the concentric moment of force production of the quadriceps [16], particularly as the angle positions are closer to the end of the knee extension. As the knee extends, the eccentric moment of force production of the hamstrings matches and overcomes the concentric moment of force production of the quadriceps. The average shape functional H:Q ratio for the knee extension across the range of motion was similar to previously limited available estimations [8,9,10,14].

To our knowledge, the only available data for the shape of functional H:Q ratio for the knee flexion across the range of motion is limited to observations in a sample of alpine skiers [17]. In the present sample of adolescent basketball players, the point of equality was not attained during the knee flexion. The observed values below the point of equality were consistent with the observations of alpine skiers [17]. This suggests that the hamstring muscles have a reduced capacity for dynamic knee joint stabilization during forceful knee flexion movements with simultaneous eccentric quadricep muscle actions [9]. Overall, the present results add to the need to consider the relations between the point of equality with functional ability and injury risk [10]. In the present study, the functional H:Q ratio for the knee in the present sample peaked at 20° and 80°, lowering between these angle positions to below 0.50 at about 0.54. Hence, the Bayesian gaam used in the present study appears to be sensitive to changes in the shape of the functional H:Q ratio for the knee flexion across the range of motion. This nonlinear trend was not observed among the alpine skiers [17]. This may reflect, at least in part, differences between the curve and functional H:Q ratio estimation method, differences in lower-body strength between the present data and the available data of adult alpine skiers, and potential bias introduced by the use of ratio standards to partition body mass on moments of force across the range of motion in the observations with the sample of alpine skiers. Ratio standards have been long noted to have theoretical and statistical limitations to remove body size influence in physiological functions [42,43], and in particular, with isokinetic moments of force [44].

## 5. Conclusions

We considered the use of gamm to describe joint-angle-specific functional H:Q ratios in the knee, using all of the available truly isokinetic data within the range. We have illustrated some of the range of possibilities that can be employed to identify and monitor individual patterns of strength development, in the present study, applied in a context of an under-15 youth basketball team. Additionally, we have considered models and techniques that are active areas of statistical research [25,26,45,46], so we do not intend to present an end-point, but rather to contribute to the understanding of the complex knee joint function and its implications for performance and injury. On-the-fly non-linear modeling allows a qualitative study of joint-angle-specific moments and functional H:Q ratios in the knee, including a consideration of individual and group characteristics, potentially providing an objective angle range of deficits to be considered for performance evaluation and medical screening, particularly with young basketball players.

## Figures and Tables

**Figure 1 medicina-55-00411-f001:**
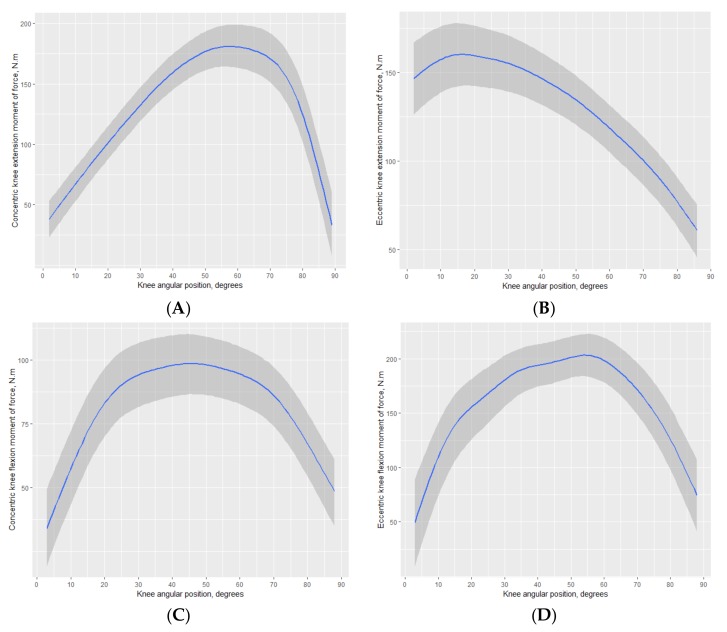
Posterior predictions for knee extension (panels (**A**,**B**)) and flexion (panels (**C**,**D**)) in both concentric and eccentric muscular actions.

**Figure 2 medicina-55-00411-f002:**
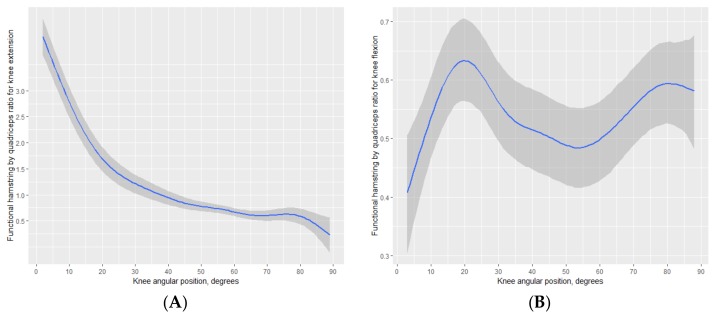
Posterior predictions for the functional hamstring to quadriceps (H:Q) ratio for the knee extension (panel (**A**)) and the knee flexion (panel (**B**)).

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
