# Peer review of "Modeling the Angle-Specific Isokinetic Hamstring to Quadriceps Ratio Using Multilevel Generalized Additive Models"

_medicina, 2019, doi:10.3390/medicina55080411_

Round 1

Reviewer 1 Report

In general, the manuscript is weakly written and extensive English changes is required. Additionally, the sentences are too long and the composition of the paragraphs is inadequate (especially; the abstract and introduction sections).

- Line 28 to 30:  This paragraph needs to be revised.

- Line 38 to 41:  This paragraph needs to be revised.

- The clarified description of subjects (Age, Weight, Height, BMI) should be mentioned in the subject section.

- Line 158 to 159 (Figure 1) and Line 167 to 168 (Figure 2): The author's utilized curve analysis for knee extension and function of H:Q ratio, but the quality of the presented figures is very low. Please, illustrates all figures again with high-quality to be clear for readers and reviewers.

- The sentences length is too long in the discussion section. Please rephrase the length of sentences again.

- Finally, can authors offer any conclusions according to their findings regarding the training consideration for basketball players?

Author Response

In general, the manuscript is weakly written and extensive English changes is required. Additionally, the sentences are too long and the composition of the paragraphs is inadequate (especially; the abstract and introduction sections).

Authors´ reply: We thank the reviewer´s comment. We revised the the text as suggested. We hope the changes improved the overall quality of the writing.  

- Line 28 to 30:  This paragraph needs to be revised.

Authors´ reply: The paragraph was revised as suggested. 

- Line 38 to 41:  This paragraph needs to be revised.

Authors´ reply: The paragraph was revised as suggested.

- The clarified description of subjects (Age, Weight, Height, BMI) should be mentioned in the subject section.

Authors´ reply: We agree with the reviewer´s comment. We added information about the characteristics of the sample as suggested.  

- Line 158 to 159 (Figure 1) and Line 167 to 168 (Figure 2): The author's utilized curve analysis for knee extension and function of H:Q ratio, but the quality of the presented figures is very low. Please, illustrates all figures again with high-quality to be clear for readers and reviewers.

Authors´ reply: We agree with the reviewer´s comment. However, we formatted the figures within the word tamplate. We added to the submission the raw figures files.

- The sentences length is too long in the discussion section. Please rephrase the length of sentences again.

Authors´ reply: The paragraph was revised the discussion as suggested.

- Finally, can authors offer any conclusions according to their findings regarding the training consideration for basketball players?

Authors´ reply: We added an applied consideration to the youth basketball sample in the conclusion as suggested.  

Author Response

We thank the reviewer´s comments. Hopefully, we have addressed each comment/question in a  <  satisfactory manner.

we added the information in the abstract as suggested. 

we changes the font in the keywords.

we added information is the first paragraph of the introduction addressing the need to contextualize within basketball players (we thank the reviewers´ comment as it improves the rationale of the manuscript)

we agree with the reviewer´s comment about the need to state more information about the standardized warm-up.

we added a short practical application of the study, in particular the benefits of on-the-fly modeling to visual interpretation of isokinetic curves and rations with athletes.

Round 2

Reviewer 2 Report

Thanks so much for addressing my comments; I have no further comments on this MS.